# Smooth Muscle Heterogeneity and Plasticity in Health and Aortic Aneurysmal Disease

**DOI:** 10.3390/ijms241411701

**Published:** 2023-07-20

**Authors:** Yunwen Hu, Zhaohua Cai, Ben He

**Affiliations:** Department of Cardiology, Shanghai Chest Hospital, Shanghai Jiao Tong University School of Medicine, Shanghai 200030, China

**Keywords:** abdominal aortic aneurysm, thoracic aortic aneurysm, VSMC, phenotypic switching

## Abstract

Vascular smooth muscle cells (VSMCs) are the predominant cell type in the medial layer of the aorta, which plays a critical role in the maintenance of aortic wall integrity. VSMCs have been suggested to have contractile and synthetic phenotypes and undergo phenotypic switching to contribute to the deteriorating aortic wall structure. Recently, the unprecedented heterogeneity and diversity of VSMCs and their complex relationship to aortic aneurysms (AAs) have been revealed by high-resolution research methods, such as lineage tracing and single-cell RNA sequencing. The aortic wall consists of VSMCs from different embryonic origins that respond unevenly to genetic defects that directly or indirectly regulate VSMC contractile phenotype. This difference predisposes to hereditary AAs in the aortic root and ascending aorta. Several VSMC phenotypes with different functions, for example, secreting VSMCs, proliferative VSMCs, mesenchymal stem cell-like VSMCs, immune-related VSMCs, proinflammatory VSMCs, senescent VSMCs, and stressed VSMCs are identified in non-hereditary AAs. The transformation of VSMCs into different phenotypes is an adaptive response to deleterious stimuli but can also trigger pathological remodeling that exacerbates the pathogenesis and development of AAs. This review is intended to contribute to the understanding of VSMC diversity in health and aneurysmal diseases. Papers that give an update on VSMC phenotype diversity in health and aneurysmal disease are summarized and recent insights on the role of VSMCs in AAs are discussed.

## 1. Introduction

The aorta is a large blood vessel consisting of three layers: intima, media, and adventitia, extending from the thorax to the abdomen. The aortic wall is eternally exposed to infinite variations of hemodynamics and neurohumoral regulators and its pathological degeneration is believed to generate aortic aneurysms (AAs) [1]. Characterized by permanent local dilation ≥ 50%, AAs occur in multiple aortic segments and are insidious, progressive, and fatal if untreated. Heritable genetic variations confer high AA risk. Alternatively, a causative gene mutation is always absent in patients with non-hereditary AAs [2,3,4,5].

As the major cellular component of aortic walls, vascular smooth muscle cells (VSMCs) ought to be dynamic to adapt to the fluctuating microenvironment and to maintain the intact structure and functionality of the aortic wall [6]. Unlike terminally differentiated cells, VSMCs can change their morphological and functional characteristics under certain conditions. VSMCs isolated from porcine aortas in sparse culture were observed to switch from a spindle to a polymorphic shape and exhibit logarithmic growth responses to mitogens [7]. Years of research have led to the contractile-synthetic binary phenotype theory of VSMCs. It is believed that VSMCs have 2 distinct but interchangeable phenotypes. The contractile VSMCs, which contribute to the maintenance of vascular tone, express high levels of contraction-related genes, including *Acta2*, *Myh11*, *Tagln*, and *Cnn1*. In contrast, synthetic VSMCs are characterized by reduced expression of contraction-related genes expression and increased cell proliferation, migration, inflammation, and extracellular matrix (ECM) synthesis [1,7,8,9,10,11,12,13,14,15,16]. According to the theory, healthy aortas contain only contractile VSMCs, and in pathological conditions, contractile VSMCs are transformed into synthetic VSMCs. Evidence that VSMCs are much more sophisticated and versatile is accumulating with recent advances in lineage tracing and single-cell RNA sequencing (scRNA-seq), challenging the binary phenotype theory [17,18].

## 2. VSMC Heterogeneity in Normal Aorta

The unique embryonic context is a source of VSMC heterogeneity. As reviewed by Majesky, fate mapping technology in developing embryos has revealed a mosaic distribution of VSMCs from different precursor sources in the aorta, which includes VSMCs derived from the secondary heart fields (SHF), cardiac neural crest (CNC), somite, and splanchnic mesoderm [19]. For instance, VSMCs of the aortic root are mainly of SHF origin, those of the ascending aorta and aortic arch originate mainly from the CNC, and VSMCs composing the descending aorta are derived from mesoderm [20,21,22]. Even the same aortic segment can consist of VSMCs of different embryonic origins. SHF-derived VSMCs were previously thought to be restricted in the aortic root, forming a suture with CNC-derived VSMCs at the transition from the aortic root and the ascending aorta. However, lineage tracing shows that SHF-derived VSMCs extend distally to the innominate artery orifice. The ascending aorta actually contains both SHF-derived and CNC-derived VSMCs, with the SHF-derived VSMCs wrapping around the outside of the CNC-derived VSMCs in a sleeve-shaped form (Figure 1A) [20,23].

According to classical theory, lineage heterogeneous VSMCs have a common developmental fate that converts to a contractile phenotype, but the scRNA-seq results reveal richer VSMC diversity than previously thought. Despite being the predominant cell type of the aorta, VSMCs have received much less attention than immune cells and endothelial cells in studies to obtain single-cell ATLAS of the aortic wall. Therefore, most of the published results simply identify VSMCs based on canonical VSMC markers without doing sub-clustering [24,25]. Nevertheless, limited data still provide a glimpse into the heterogeneity of VSMCs in normal aorta walls from a transcriptome perspective. In one study, a well-performing classifier based on the VSMC transcriptome obtained by scRNA-seq was constructed to discriminate the regional identity of VSMCs (AUC = 0.9967), suggesting that aortic segmental identity is preserved in individual VSMCs [26]. Even within aortic segments, VSMCs are heterogeneous and have significant intercellular variation in genes associated with contraction, inflammation, local adhesion, migration, and proliferation [26]. In addition, sc-RNAseq revealed six VSMC subpopulations distributed throughout the aorta, and no segment-specific subpopulation was identified [27].

Notably, Lina and her team innovatively combined lineage tracing, fluorescence-activated cell sorting (FACS), and scRNA-seq to identify and isolate a small subset of Sca1-expressing VSMCs from normal mouse thoracic aortas [26]. Sca1^+^ VSMCs express reduced contractile signatures and show enrichment in genes associated with migration, proliferation, synthesis, and secretion compared to Sca1^−^ VSMCs. Given that Sca1 has been widely used as a biomarker of stem cells and cell stemness, it is possible that Sca1^+^ VSMCs represent a de-differentiated state [26,28]. Although Sca1^+^ VSMCs are rare, they are widely distributed throughout the aorta. Furthermore, Sca1 expression was detected in all VSMC subgroups with different contractile identities. The above results demonstrate a ubiquitous distribution of Sca1^+^ VSMCs [27]. Mature VSMC can express detectable levels of Sca1 in two different contexts: those in or about to undergo phenotypic transformation and those newly differentiated or recruited from vascular stem cell pools [29,30,31,32]. Because scRNA-seq can only capture transient transcriptomic data, we cannot determine whether Sca1 expression in mature VSMCs is transient. It is also difficult to label and follow the fate of Sca1^+^ VSMCs due to the scarcity of these cells. Nonetheless, these results are significant as it indicates that VSMCs in healthy aortic walls are not static.

The heterogeneity of VSMCs may be a physiological adaptation. Hemodynamics vary in different segments of the aorta. Blood velocity in the ascending aorta and aortic arch is faster than in the abdominal aorta, causing greater vessel wall impact [33,34]. Due to the anatomical curvature, the blood flow in the ascending aorta, especially in the aortic arch, changes drastically, resulting in vortex flow and increased demands on the strength and self-healing ability of the vascular wall [35,36,37]. Vim-expressing VSMCs enriched for genes associated with shear stress, atherogenesis, WNT, and MAPK pathways populate the thoracic aorta and aortic arch, representing a group of flow-shock-adapted VSMCs that contribute to higher elastin/collagen content [27]. Interestingly, the regional distribution of VSMCs from different embryologic backgrounds is consistent with hemodynamics, with a progressive increase in the proportion of CNC-derived VSMCs along the ascending aorta to the aortic arch suggesting that the composition of the VSMCs of distinct lineage influences the intensity of the arterial wall. Indeed, VSMCs of different embryonic origins are phenotypically and functionally distinct, with different proliferation and protein expression patterns in response to cytokines [38]. Stimulation with wound repair factor transforming growth factor-beta 1 (TGF-β1) or platelet-derived growth factor BB (PDGF-BB) significantly enhanced the biosynthesis of ECM components such as Col1A1 in CNC-derived VSMCs and promoted cell proliferation by increasing c-myb expression, but there was no effect on VSMCs of mesodermal origin, suggesting that embryonic lineage is a major source of aortic VSMC heterogeneity [39,40,41].

## 3. VSMC Phenotypic Diversity in Hereditary AAs

The susceptibility to aortopathy varies between aortic segments. The ascending aorta is less prone to atherosclerosis than the abdominal aorta, and even thoracoabdominal aortic graft exchange does not alter atherosclerotic propensity, suggesting that the intrinsic nature of the arterial wall influences vulnerability to arterial disease [42,43,44]. Segment-specific distribution of heterogeneous VSMCs likely contributes significantly to aortic disease characteristics. Further support for this view is the fact that the aortic arch, which is composed of CNC-derived VSMCs, is the segment of the thoracic aorta that is more susceptible to calcification [45,46]. As VSMCs of different embryonic origins have heterogeneous patterns in response to environmental cues, genetic mutations that can affect intracellular signaling could also have heterogeneous effects on them. Patients with genetic defects in focal adhesion mediators or components of the TGF-β signaling cascade are predisposed to thoracic aortic aneurysms (TAAs) with manifestations of syndromic TAAs such as Marfan syndrome (MFS), Loeys–Dietz syndrome (LDS), or TAAs alone [47,48,49,50,51,52,53,54,55]. Importantly, almost all AAs in LDS and MFS patients occur in the aortic root and ascending arteries, and dissections often arise in the ascending aorta (Figure 1B), suggesting that heterogeneous lineage plasticity of VSMCs determines the pathological characteristics of hereditary aneurysms [56,57].

### 3.1. VSMC Phenotypic Diversity in LDS

LDS is an aneurysm-predisposing disease caused by defects in the canonical positive regulators of TGF-β, which regulate the differentiation and maturation of VSMCs [58,59]. Heterozygous loss-of-function mutations in genes encoding multiple components of the TGF-β signaling pathway, including the ligands (TGFB2/3), transforming growth factor-β receptor types I and II (TGFBR1 and TGFBR2) and downstream effectors (SMAD2/3), cause abnormalities in the function of VSMCs (Figure 1C).

Approximately 20–25% of patients with LDS have mutations in the *TGFBR1* genes [60]. The embryonic background of the VSMCs determines the effect of this genetic defect. SHF-derived VSMCs, but not CNC-derived VSMCs, exhibited impaired Smad2/3 activation, increased angiotensin II type 1 receptor (Agtr1a), and TGF-β ligands expression in the LDS model carrying heterozygous Tgfbr1 inactivation [61]. A human-induced pluripotent stem cell (hiPSC) model derived from an LDS patient family with the *TGFBR1^A230T^* variant also detected disruption of SMAD3 and AKT activation and significantly reduced contractile transcript and protein levels in SHF-derived VSMCs, but not in CNC-derived VSMCs [62]. Single-cell transcriptomic data revealed molecular similarities between SHF-derived VSMCs with the *TGFBR1^A230T^* variant and SHF-derived VSMCs with a loss-of-function SMAD3 mutation. Additionally, pharmacological activation of the intracellular SMAD2/3 signaling cascade can rescue contractile gene expression and contractile function in TGFBR1 mutant SHF-derived VSMCs [62]. The above results indicate that TGFBR1 maintains the contractile phenotype of SHF-derived VSMCs through the activation of SMAD3. Smad3 interacts with multiple VSMC-specific promoters to facilitate a contractile phenotype [63]. The deficiency of SMAD3 in SHF-derived VSMCs induced from hiPSC significantly impaired canonical TGF-β signaling, decreased VSMC key regulators and markers, and increased collagen expression, suggesting a transition towards a less contractile phenotype [64]. Disruption of pathways related to ECM organization and downregulation of key focal adhesion components, including integrins and anchoring protein, were also found in *Smad3^−/−^* VSMCs derived from SHF [65]. Furthermore, selective knockdown of Smad3 in VSMC resulted in an up-regulation of pro-pathogenic factors such as thrombospondin-1, angiotensin-converting enzyme, and pro-inflammatory mediators in VSMC subsets located in the aortic root (mainly derived from SHF), leading to greater dilation and histologic abnormalities [65]. Reduced expression levels of contractile genes and dissociation from ECM components of SHF-derived VSMCs explain the reduction in vascular wall tone and local dilation in the aortic root [66]. CNC-derived VSMCs, however, were less affected by impaired Tgfbr1-Smad3 pathway and can always maintain the contractile phenotype by upregulating p-Smad2, suggesting a bypass might be exploited [61,62,64]. The inconsistency of phenotypic changes of CNC and SHF-VSMCs results in the decreased intensity of the interface between them in the ascending aorta, making it susceptible to type A aortic dissection (AD) [67]. Remarkably, although CNC-derived VSMCs retained significant VSMC markers in the presence of Tgfbr1-Smad3 dysregulation, elastin expression was significantly decreased, indicating phenotypic changes at the transcriptome level.

Approximately 55–60% of LDS patients carry *Tgfbr2* mutations [60]. The effect of Tgfbr2-mediated signaling activation on VSMC gene expression depends on the location of VSMCs in the aorta [68]. Smooth muscle specific *Tgfbr2* deficiency significantly enhances the progression of ascending but not abdominal aortic pathology, including increased intramural hemorrhage, medial thinning, and epicardial thickening, and damage was more pronounced outside of the media layer than inside [69,70]. Although rigorous experimental evidence such as lineage-stratified sc-RNAseq and lineage tracing is required, it is speculated that VSMCs of different embryonic origins respond heterogeneously to Tgfbr2 deficiency. Whether *Tgfbr2* exerts a facilitative or inhibitory effect on the differentiation of neural crest cells (NCCs) into VSMCs remains unclear. TGF-β signaling is involved in NCCs migration and differentiation and germline knockout of *Tgfbr2* in VSMCs leads to extensive and lethal cardiovascular malformation in mouse embryos [68,71,72,73]. *Tgfbr2* deficiency was found to hinder NCC-to-VSMC differentiation, resulting in a deficiency of CNC-derived VSMCs in the aorta [73,74]. However, recent studies have shown that *Tgfbr2* deficiency does not impede the VSMC fate of NCCs and may even lead to premature differentiation of NCCs into VSMCs [68,75,76]. Smad2 is confirmed as an essential regulator in progenitor-specific VSMC development and physiological differences between CNC- and mesoderm-derived VSMCs [72,77,78]. Although in vitro studies found Smad2 functionally intact in *Tgfbr2*-deficient VSMCs, animal models demonstrated reduced p-Smad2 levels [70,77,79]. Embryonic *Smad2* knockdown in NCCs reduces the number of CNC-derived VSMCs and produces a damaged, fragmented elastin lamina in the media of the ascending aorta [70,77,80]. Strikingly, postnatal knockdown of Tgfbr2 in VSMCs, while reducing Tgfb signaling, has little effect on contractile gene expression levels and contractile function in VSMCs, but rapidly induces elastin fragmentation and ascending thoracic aorta disruption (ATAD) [70,76]. Although conditional knockout of *Tgfbr2* at 11 months of age failed to induce TAAD, the ascending aorta developed pathological changes similar to those observed in elastin-deficient (*ELN^+/−^*) mice [81,82]. Given that the deposition of the elastin membrane occurs only during early life and *TGFBR2* mutation was associated with a significantly higher risk of aortic events with childhood onset [83]. CNC-derived VSMCs are speculated to involve in the assembly and deposition of elastin lamina during embryonic development and are primarily responsible for the maintenance of elastin lamina integrity in adulthood [84,85]. Since the elastin lamina is a key component in maintaining the elasticity of the vessel wall, disruption of this structure impairs the absorption and buffering of kinetic energy from the blood flow by the aortic wall, resulting in a greater and more direct kinetic energy impact on the aorta. This provides a plausible explanation for the higher probability of developing ATAD and for the smaller mean arterial diameter at the time of dissection in LDS patients with Tgfbr2 mutations compared to LDS patients with Tgfbr1 mutations [86,87,88].

### 3.2. VSMC Phenotypic Diversity in MFS

FBN1 gene mutations predispose Marfan syndrome (MFS) patients to TAA at a young age and premature death from catastrophic aneurysm rupture or dissection [56]. The *FBN1* gene does not directly regulate the contraction-related phenotype of VSMC, but its coding product fibrillin-1 is an important component of the ECM and acts to mediate the attachment of VSMCs to elastin laminae [89]. Fibrillin-1, the encoded product of FBN1, is an important component of the ECM that mediates the attachment of VSMCs to elastin laminae, the loss of which would disrupt mechanotransduction and result in detached VSMCs with altered morphology, reduced contractile signatures, and upregulation of several ECM elements and elastolysis mediators indicating cellular plasticity. (Figure 1C) [90].

MFS VSMCs derived from iPSCs showed a lineage-specific protein profile. Compared to CNC-derived VSMCs, SHF-derived VSMCs have a proteome with less VSMC identity (*TAGLN*), but increased levels of focal adhesion components (integrin αV, fibronectin) and ECM remodeling molecules (collagen type 1, MMP2) [91]. As a novel protein marker associated with MFS aneurysms, uPARAP, an endocytic receptor responsible for receptor-mediated internalization of collagen for degradation, was specifically downregulated in SHF-derived VSMCs [91,92]. Recent work has shown that uPARAP also acts as an endocytic receptor for thrombospondin-1 (TSP-1), which has been reported to promote AAs by increasing aortic inflammation and TGF-β activity [93,94,95,96,97]. Single-cell transcriptomics provides evidence for lineage-specific VSMC plasticity in MFS aortic aneurysms. Albert and colleagues identified a group of modulated VSMCs (modVSMCs) in aortic aneurysm tissue from *Fbn1^C1041G/+^* mice with a transcriptome similar to that of VSMC-lineage-derived fibromyocytes in atheroma [98]. The combination of lineage tracing and scRNA-seq uncovered that SHF-derived VSMCs and CNC-derived VSMCs contribute equally to modVSMCs [99]. However, a lineage-stratified perspective reveals heterogeneous transcriptomes of VSMCs from different embryologic origins. SHF-derived VSMCs increased biosynthesis of collagens and small leucine-rich proteoglycans (SLRPs), contributing to the excessive deposition of ECM in MFS aortic aneurysms [98,99]. In contrast, CNC-derived VSMCs showed an osteochondrogenic phenotype, which is relevant for the medial calcification of the aorta [45,100]. In particular, few studies suggested that MFS aortic aneurysms have a hypercontractile phenotype with significant cytoskeletal density, upregulation of focal adhesions, and some contractile regulators presumably resulting from overactivation of TGF-β signaling [101,102]. Because the current scRNA-seq analyses focus on subdividing VSMCs with downregulated contractile markers, the hypercontractile VSMCs are likely to be buried in the normal contractile VSMC subgroup. Taking into account the lineage-specific response pattern of VSMCs to TGF-β, hypercontractile VSMCs, if they exist, are more possible to be derived from CNC-derived VSMCs.

## 4. VSMC Phenotypic Diversity in Non-Hereditary AAs

Apart from a few that can be explained by causative genetic mutations, most sporadic patients do not have genetic defects and these AAs are called non-hereditary AAs [103,104,105]. The etiology of non-hereditary AAs is still enigmatic and previous studies have shown that risk factors for non-hereditary AAs, such as hypertension, aging, and hyperlipidemia, transform contractile VSMCs into synthetic VSMCs [105,106,107,108,109,110]. However, the definition of synthetic VSMCs is ambiguous and recent studies demonstrate that they are actually conglomerates of several VSMC phenotypes with different functions (Figure 2 and Table 1).

### 4.1. VSMC Diversity in TAAs

#### 4.1.1. Secreting VSMCs

Although investigators annotate VSMCs differently, several studies have found a subcluster of VSMCs positioned between VSMCs and fibroblasts in both UMAP and tSNE projections [111,112,113,119]. It maintains or moderately reduces the expression of contraction-related genes, but upregulates genes enriched in cell adhesion and ECM remodeling [15,16]. This subcluster is referred to as fibromyocytes or secreting VSMCs due to their properties of both contractile VSMCs and fibroblasts. Unlike “synthetic” VSMCs, secreting VSMCs do not activate cyclin and pro-inflammatory gene expression. Although rare in healthy aortic walls, secreting VSMCs significantly increased their proportion in aneurysmal aorta, further reducing VSMC identity markers such as *ACTA2* and *MYH11* and expressing ECM genes such as *SERPINE1* and *FN1* [119]. *VCAN*, a gene encoding Versican, a chondroitin sulfate proteoglycan composing the ECM, was identified as a marker of secreting VSMCs. Trajectory analysis of VCAN^+^ VSMCs further supports the phenotypic transition from contractile VSMCs to fibroblasts [113,120]. It should be noted that the transition to secreting VSMCs is common in several aortic diseases. In atheroma, VSMC-traced scRNA-seq confirmed that the fibromyocytes are derived from contractile VSMCs, and they can be clustered with modVSMCs in MFS [98,121,122]. In addition, secreting VSMCs in hypertension-induced TAAs also have similar transcriptomes to modVSMCs in MFS [123]. Fibromyocytes contribute to the formation of the fibrous cap and are regarded as a protective factor in atherosclerosis [76,77] A protecting role of secreting VSMCs has also been suggested in TAAs, as they are associated with an increase in aortic wall thickness [114]. Interestingly, GO analysis showed that genes enriched in cardiac muscle cell development were highly expressed in secreting VSMCs which implies that secreting VSMCs may originate from a group of VSMCs regressed to an embryonic state and that HDAC9 and TGF-β pathways are involved in this change [119].

#### 4.1.2. Proliferating VSMCs

VSMC proliferation is a hallmark of occlusive aortic diseases such as restenosis after angioplasty or stenting [124]. Upon injury to the vascular wall, the quiescent, differentiated VSMCs have been reported to be activated to a proliferative phenotype at low frequencies and such oligoclonal contribution of VSMCs is a feature of obstructive arterial disease [125,126,127,128]. Fate mapping analyses in TAA models induced by disruption of the TGF-β pathway also revealed oligoclonal expanding of the VSMCs [115,116]. Enhanced VSMC proliferation was demonstrated in sporadic TAAs [111,112,114]. In human TAA samples, sc-RNAseq identified a subpopulation of VSMCs characterized by high levels of cyclin expression and low levels of cell adhesion, which is highly suggestive of ongoing cell proliferation and was annotated as proliferating VSMCs [111]. Despite expressing some canonical synthetic marker genes such as *MGP*, *TPM4*, and *MYH10*, the proliferating VSMCs do not upregulate ECM and inflammatory genes, which is different from synthetic VSMCs [111]. In particular, the moderately reduced expression of contraction-related genes strongly suggests that this phenotype is activated from quiescent, mature VSMCs [111]. These results indicate that some mature VSMCs in the aortic wall have proliferative potential and can be activated to a proliferating state in TAAs. However, the identity of contractile VSMCs seems to be lost to varying degrees during the initiation of cell proliferation. Re-analysis of the sc-RNAseq data in reference [112] revealed the presence of mesenchymal-like VSMCs with a characteristic expression of ACTA2, MYH11, CD34, and PDGFRA and with a minor role in the communication network [112]. Fate mapping analyses have observed mesenchymal cells (osteoblasts, chondrocytes, adipocytes, and macrophages) derived from clone-expanded VSMCs in the adventitia of mice with TAAs, suggesting the pluripotency of proliferating VSMCs [115,116]. The aforementioned results suggest that the proliferating VSMCs represent a group of partially dedifferentiated VSMCs. However, it is still uncertain whether the proliferating VSMCs are the result of phenotypic transformation or clonal selection.

#### 4.1.3. Immune-Related VSMCs

In the context of atherosclerosis, VSMCs have been shown to have the potential to switch to monocyte/macrophage lineages, implicating them in the modulation of inflammation in the vascular wall [128,129,130,131]. Although medial degeneration has been confirmed as the most common pathological manifestation of TAAs, aortitis, and atherosclerosis remain the main histopathological substrates in 1/3 of TAA patients [132,133]. The prevalence of inflammation-related histologic changes underscores the pivotal role of the inflammatory response in the pathophysiology of TAAs. VSMCs that exhibit similar characteristics of monocyte/macrophage and T-lymphocyte are identified in TAAs and referred to as immune-related VSMCs [111,112,117]. Additionally, an intermediate state, which expresses interferon-induced genes, has the potency to switch to T-cell-like VSMCs or macrophage-like VSMCs, suggesting that VSMCs are activated by inflammation [111]. Bioinformatics analysis revealed that immune-related VSMCs exert immunomodulatory functions through rich and strong interactions with immune cells [112]. T-cell-like VSMCs recruit immune cells mainly through the CXCL12-CXCR4 pair. Macrophage-like VSMCs in TAAs can communicate with all infiltrating immune cells through PTN and galectin, which are related to the suppressive immune microenvironment in malignancies [134,135,136]. The immunosuppressive function of macrophage-like VSMCs may explain why the inflammatory response in Ang II-induced TAAs is not as strong as in AAAs. Notably, macrophage-like VSMCs were detected in the adventitia and pseudolumen, accumulating in thrombotic/hemorrhagic regions and upregulating phagocytic and lysosomal pathways, which suggests that macrophage-like VSMCs have vigorous phagocytic activity [116]. In combination with the promotion of macrophage-mediated efferocytosis for the prevention and treatment of vascular disease, macrophage-like VSMCs in TAAs are hypothesized to play an immune clearance role by removing cells and ECM from injured sites in preparation for clones of proliferative VSMCs and ECM newly synthesized by secreting VSMCs to replace the injured tissue [137,138]. In addition, macrophage-like VSMCs, unlike terminally differentiated peripheral blood-derived macrophages, were recently found to be plastic and capable of changing their lineage characteristics toward fibroblasts and pericytes in atheroma, supporting a role for them in maintaining vessel wall stability by exerting a repair mechanism [131].

### 4.2. VSMC Diversity in AAAs

#### 4.2.1. Proinflammatory VSMCs

AAA is considered to be a chronic inflammatory disease. Pro-inflammatory VSMCs defined by their function to promote aortic wall inflammation are identified. They have a proinflammatory transcriptome characterized by the expression of proinflammatory cytokines (IL-1ß, IL-6, CCL2, CCL5, and TNFα), chemokines (CXCL10 and ICAM1), and high expression of ECM-degrading components such as MMP2/9. Aortic wall inflammation can damage VSMCs, activate the expression of inflammation-related genes through innate immune pathways, and further promote vascular wall inflammation, forming a vicious cycle [139,140,141]. VSMCs expressing macrophage markers are also present in AAAs. [27,118]. In contrast to their immunomodulatory counterparts in TAAs, macrophage marker-expressing VSMCs in AAAs play an active role in inflammation and ECM degradation. Studies have shown that not only do these cells have high expression of pro-inflammatory cytokines and chemokines, but they also have impaired phagocytosis and efferocytosis and are unable to remove apoptotic VSMCs in a timely manner, perpetuating the pro-inflammatory effects of injured VSMCs [142,143,144]. Mechanically, disturbances in intracellular calcium ion balance drive the conversion of VSMCs into pro-inflammatory macrophages-like phenotype. Downregulation of the sarcoplasmic reticulum calcium-pumping ATPase was found in macrophage-like VSMCs in AAAs suggesting impaired intracellular calcium homeostasis [118]. Consistently, activation of P2Y2 receptors and TRPV4 channels on VSMCs by Panx1-mediated ATP release from endothelial cells destabilized intracellular Ca^2+^ homeostasis, stimulated proinflammatory cytokine secretion, and facilitated increased MMP2 activity [145]. Notably, antibody-dependent adaptive immune responses are involved. For example, IgG immune complex can polarize VSMCs from a contractile phenotype to a pro-inflammatory phenotype expressing M1 macrophage markers through FcγR in a TLR4-dependent manner [144,146].

#### 4.2.2. Senescent VSMCs

Chronic inflammation of the vessel wall is bound to cause the death of VSMCs, and the depletion of contractile VSMCs is a hallmark of AAAs. The scRNA-seq analyses failed to identify any proliferating VSMCs in the AAA specimens, indicating the failure of the VSMCs to activate into a proliferative phenotype. Nevertheless, a group of VSMCs that specifically express high levels of GATA6 and undergo glycolytic reprogramming has been identified [118]. GATA6 plays a multifaceted role in regulating VSMC phenotype. GATA6 has been implicated in the differentiation of VSMCs, as studies have shown that specific knockout of GATA6 in VSMCs can activate the expression of synthetic markers [147,148]. Recently, however, it has been shown that GATA6 overexpression promotes TGF-β and MAPK signaling, resulting in increased proliferative and migratory capacity in VSMCs [149]. Notably, GATA6 is involved in the aging of mesenchymal stem cells by regulating the Hedgehog and FOXP1 signaling pathways [150]. Based on the increased expression of age-related metallothionein proteins, this VSMC subgroup is speculated to be a subset of senescent dedifferentiated VSMCs [151]. It is reasonable to speculate that senescence and loss of stemness of VSMCs are critical in the pathogenesis of AAAs, given the therapeutic efficacy of mesenchymal cell intervention for AAAs in preclinical studies [152]. It is also notable that He et al. identified a group of sparse VSMCs in patients with ATAD. These VSMCs simultaneously decreased the levels of VSMC markers, cell adhesion, the calcium-mediated signaling pathway, the cGMP metabolism, as well as the translation and RNA metabolism process. Due to the extremely low differentiation potential and the high expression of metallothionein, we speculate that these are also senescent VSMCs, although they are not annotated by the author [117].

#### 4.2.3. Stressed VSMCs

Stress events such as oxidative stress, mitochondrial dysfunction, and unfolded protein response have been shown to alter the VSMC phenotype toward pro-inflammation and proliferation [153,154,155,156]. A group of VSMCs with transcriptomic features similar to contractile VSMCs, except a characteristically high expression of stress-related genes (*FOS*, *ATF3*, *JUN*, and *HSPB8*), were identified in TAAs and annotated as stressed VSMCs [111]. VSMCs with contractile VSMC transcriptome and high *FOS*, *Jun*, *Klf2*, and *ATF3* expression were also identified in elastase-induced AAAs [118]. Due to the relatively high expression of genes associated with proliferation, they were annotated as proliferative VSMCs by the other. However, the actual proliferative capacity of this group of cells is uncertain because they express a high level of mitogen-activated protein kinase phosphatase-1 (Dusp1), which inhibits proliferation by suppressing MAPK activity. In addition, upon elastase challenge, this group of cells exhibits an inflammation-activated profile, with significantly upregulated expression levels of pro-inflammatory cytokines (Cxcl2 and Ccl2). Therefore, it would be more appropriate to annotate them as stressed VSMCs [118]. Utilization of the transcription-dependent adaptive response implies that cells are subjected to prolonged stimuli, suggesting that they are at the crossroads of cell fate: necrosis/apoptosis or adaptive compensation. Given the significant reduction of medial VSMCs and the presence of a senescent phenotype in AAAs, stressed VSMCs in the abdominal aorta tend to go to extinction compared to their counterparts in the thoracic aorta, but the reasons for this remain to be elucidated.

## 5. Role of VSMC Phenotypic Diversity in AAs

In clinical samples from patients with AAs, changes in VSMC phenotype affect hemostasis of the arterial wall, and VSMC plasticity has been correlated with disease severity. Such studies, however, represent only the final stage of the disease. Thus, whether VSMC heterogeneity and plasticity play a causative or compensatory role in AAs remains controversial. Animal models are tools for studying VSMC pathogenesis and phenotypic disruption of VSMCs can be inferred to cause hereditary aortic aneurysms from genetically engineered models [62,66,157]. VSMCs of different lineages respond differently to the same genetic mutation, and SHF-derived VSMCs are more phenotypically vulnerable than CNC-derived VSMCs. A recent study reported that the spatial specificity of TAAs has been linked to the downregulation of LRP and TGF-β signaling in SHF-derived VSMCs in response to Ang II stimulation [158]. Epigenetic mechanisms may account for the phenomenon that VSMCs of different lineages respond heterogeneously to the same signaling dysregulation. Aneurysmal lesions of MFS are characterized by dysregulated miRNA networks and gene expression profiles [159]. Epigenetic regulatory networks also have an impact on the response of VSMCs to pathogenic signals such as Ang II and TGF-β [160,161,162]. However, studies to date have mainly focused on epigenetic dysregulation in pathological states, and few have compared differences between VSMCs in different parts of the aorta in physiological or pathological states from an epigenetic perspective [163].

In contrast to hereditary AA, sporadic AA is a multifactorial disease that poses difficulties in the construction of animal models. At present, no animal model perfectly mimics human non-genetic AAs [164,165,166]. Results from non-hereditary models of AA support to some extent, the compensatory role of VSMC phenotypic transformation. To induce AA in WT mice, a minimum of two risk factors as well as sufficient challenge time are required. Challenging the aorta with a single risk factor such as Ang II infusion, hypertension, or hypercholesterolemia alone can elicit phenotypic changes of VSMCs but is not insufficient to induce AAs [105,106,107,108,109,110]. Extreme agents such as calcium chloride and elastase, which directly destroy the aortic wall, do induce AAs [167,168,169]. However, such “one-hit” models failed to reproduce the progression of human AAs, indicating that a stable adaptive state can be established [118,165]. Several VSMC subsets are thought to participate in compensatory responses to maintain aortic wall integrity. Collagen and proteoglycan produced by secreting VSMCs strengthen the aortic wall and prevent it from rupturing. The immune-related VSMCs suppress the immune response and assist in the removal of damaged cells from the aortic wall. The proliferative and MSC-like VSMCs are able to replicate and replace the damaged VSMCs that have been depleted by the immune cells. Even in inherited AA caused by genetic mutations that damage the “elastin-VSMC contractile unit” represented by MFS, the compensatory effect of the VSMC was observed [53,66,170]. In MFS aortic aneurysms, compensatory manifestations of impaired aortic wall force generation include the transition of contractile VSMCs to modVSMCs with increased ECM production capabilities and hypercontractile VSMCs and thickened aortic walls.

Despite the potential protective effects of the phenotypic alteration of VSMCs, the development and progression of sporadic AAs are indicative of maladaptation. We propose that the sustained presence of noxious stimuli causes a prolonged phenotypic transformation of VSMCs, which causes pathological remodeling of the aortic wall and eventually leads to non-hereditary AAs. Proteoglycan and collagen accumulation prevent aortic dissection and rupture, however, overaccumulation of proteoglycans and collagen results in stiffening of the aortic wall which disrupts the phenotypic change of VSMCs [171,172]. Elevated ECM stiffness has been reported to polarize contractile VSMCs toward proinflammatory phenotypes through mechanotransduction [173,174,175,176]. Chronic vascular wall inflammation then leads to persistent programmed death of VSMCs, ultimately exhausting the proliferative potential of VSMCs. Ideally, the transition to proliferative VSMCs is reversible and proliferative VSMCs can re-differentiate into mature, contractile VSMCs. Nevertheless, if the stimuli are prolonged, epigenetic changes can occur, making the change irreversible [16,177,178]. In addition, the precise microenvironment that regulates VSMC phenotype is perturbed in the pathologically remodeled aortic wall, resulting in disrupted VSMC differentiation. For example, an aberrant equilibrium between ECM stiffness and growth factors in aneurysmal tissue will impair the differentiation of MSC-like cells toward contractile VSMCs [179,180]. Moreover, high levels of TGF-β activation in AAs are thought to over-activate nonclassical TGF-β signaling in VSMCs, which breaks the normal physiological balance and results in AAs [181]. Chronic activations of mTOR signaling would elevate the expression and activity of MMPs derived from VSMCs, resulting in irreversible destruction of the elastin membrane and VSMC microcalcifications [100,182,183].

## 6. Conclusions and Future Perspectives

The arterial wall is composed of VSMCs of distinct origins that exhibit different transcriptomic profiles and dynamic phenotypes to adapt to changing environments. Their unique distribution and lineage-specific biological behaviors in the presence of AA-causing gene mutations determine the segmental vulnerability of hereditary AAs. VSMCs are highly plastic and can transform into cellular phenotypes with distinct functions in non-hereditary AAs. Reversible phenotypic transformation might play a beneficial role in maintaining the integrity of the aortic wall. However, chronic exposure to risk factors leads to irreversible phenotypic changes in VSMCs and to pathological remodeling of the aortic wall, which contributes to the pathogenesis and development of non-hereditary aortic aneurysms.

Lineage tracing and sc-RNAseq have allowed detailed phenotyping and revealed a high degree of diversity of VSMCs in AAs. Specific patterns of modulating key pathways that regulate contractile phenotypic transformation are observed in VSMCs from different embryological backgrounds. Heterogeneous responses can be induced by genetic defects or environmental cues, as long as they can affect any part of the pathway. Thus, part of the intrinsic nature of aortic segments can be attributed to the lineage-specific VSMC response to perturbation. For instance, the unique composition of VSMCs derived from the splanchnic mesoderm contributes, to some extent, to the inflammatory tendency of the abdominal aorta. However, little is currently known about VSMCs other than those derived from SHF and CNC, and the importance of lineage stratification needs to be emphasized. In addition, comprehensive and in-depth studies are needed to depict the molecular regulatory profiles specific to different lineages of VSMCs.

VSMCs can switch to a variety of functionally distinct phenotypes. How VSMC plasticity contributes to the pathogenesis and progression of AAs remains to be fully elucidated. It is important to note that the transcriptome level heterogeneity obtained by sc-RNAseq might be a reflection of transient cell states rather than stable phenotypes. Our conclusions are mainly based on the results of sc-RNAseq, but technical limitations must be considered to fully evaluate our results. For example, heterogeneity at the transcriptome level may reflect only transient cellular states and not necessarily stable cellular phenotypes. To demonstrate the existence of specific dysfunctional phenotypes, the next step will be to attempt to isolate and induce stable VSMC subsets revealed by sc-RNAseq and summarize their ultra-structural features, which are currently not available. Furthermore, sc-RNAseq can only capture the transcriptome profile of specific developmental stages. Comparison and evaluation of VSMC phenotypes at different stages of AA development will help us to understand the pattern of VSMC phenotypic changes and their impact on aortic wall integrity which we believe that will put an end to the controversy about the causal/successive relationship between VSMC plasticity and the development of AAs and provide a theoretical basis for VSMC phenotype modulation as a potential pharmacological intervention.

## Figures and Tables

**Figure 1 ijms-24-11701-f001:**
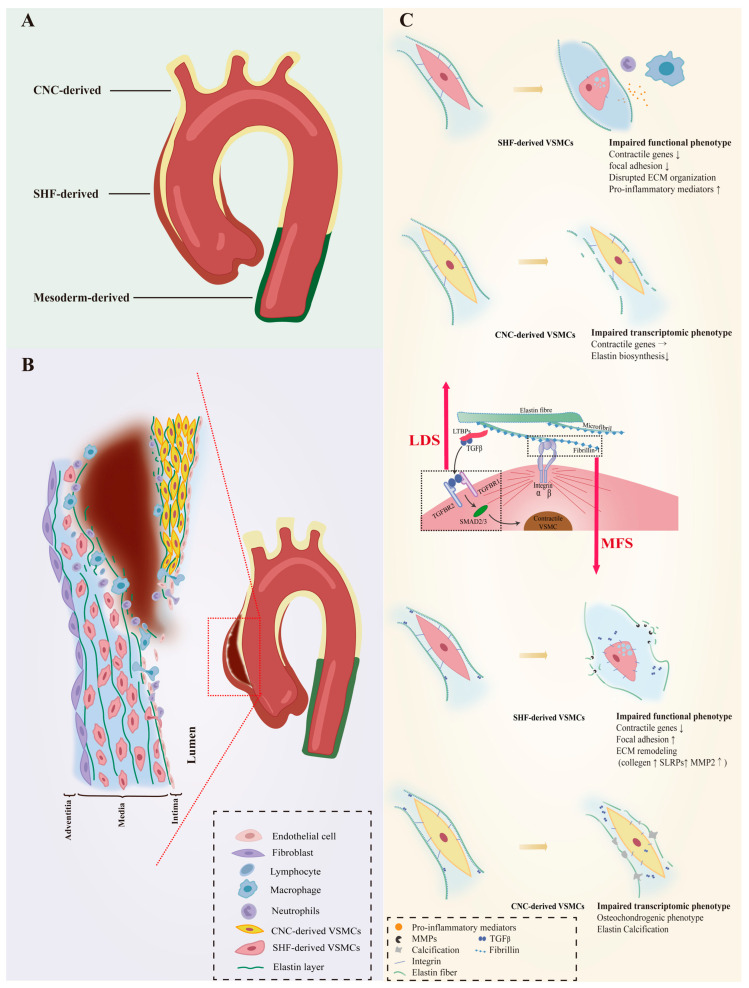
VSMC phenotypic diversity in hereditary AAs. The aortic wall consists of VSMCs from different embryonic origins that respond unevenly to genetic defects that directly or indirectly regulate VSMC contractile phenotype, which results in heterogeneous susceptibility to aortic diseases. (**A**) provides an overview of the origin of VSMCs in the aortic wall. The ascending aorta is composed of VSMCs derived from SHF and CNC. The former wraps outside of the latter in a “sleeve-shaped” form, However, mesoderm-derived VSMCs predominate in the abdominal aorta. (**B**) depicts a plausible hypothesis that the ascending aorta is susceptible to AAs and ADs when genetic defects are present. VSMCs with distinct embryonic backgrounds show different responses to signaling perturbations. Compared to CNC-derived VSMCs, SHF-derived VSMCs are more prone to lose their contractile phenotype and cause pathological ECM remodeling. This leads to a reduction in the strength of the aortic wall, making it susceptible to swelling. Moreover, the turbulent blood flow can easily cause tearing along the interface between the 2 VSMCs, leading to the occurrence of ADs in the presence of damaged endothelium. (**C**) shows the dysfunctional phenotypes of VSMCs from different origins with genetic defects. Fibrillin-1 is the primary protein found in the microfibril extensions of elastic lamellae, which are attached to focal adhesions on the surface of VSMCs. It is also associated with TGF-β binding proteins (LTBP), which are involved in the TGF-β signaling pathway. Genetic defects in fibrillin-1 (MFS) and defects in the TGF-β signaling pathway (LDS) affect the phenotype of both VSMCs, with SHF-derived VSMCs being more susceptible to loss of the contractile phenotype. The direction of arrows indicates the trend of gene expression levels: upward represent upregulation, horizontal indicate stable expression levels, and downward indicate downregulation. Vascular smooth muscle cells, VSMCs; Secondary heart field, SHF; Cardiac neural crest, CNC; Extracellular matrix, ECM; Aortic aneurysms, AAs; Aortic dissections, ADs; Loeys-Dietz syndrome, LDS; Marfan syndrome, MFS.

**Figure 2 ijms-24-11701-f002:**
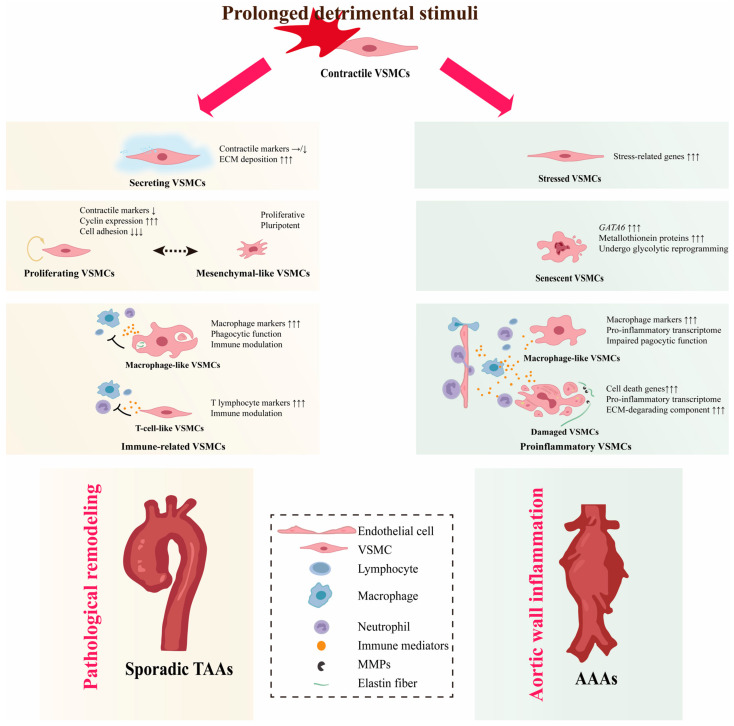
VSMC phenotypic diversity in non-hereditary AAs. VSMCs transform into different phenotypes and show a high degree of plasticity when exposed to detrimental stimuli. In the thoracic aorta, secreting VSMCs cause deposition of ECM components. Proliferative VSMCs undergo active proliferation and immune-related VSMCs will exert immune clearance and immunomodulatory effects. In the long term, these non-contractile phenotypes can result in pathological remodeling and, ultimately, to TAAs. In the abdominal aorta, stressed VSMCs are standing at the at the crossroads of necrosis/apoptosis or compensation. Pro-inflammatory VSMCs maintain and exacerbate vessel wall inflammation, eventually exhausting proliferative VSMCs and triggering a senescent phenotype, which leads to VSMC depletion and disruption of aortic wall integrity, triggering AAAs. Arrows indicates the gene expression changes: the number of arrows indicates the magnitude of expression change, and the direction of arrows indicates the trend, upward represent upregulation, horizontal indicate stable expression levels, and downward indicate downregulation1.

**Table 1 ijms-24-11701-t001:** Studies of VSMC phenotypes in non-hereditary AAs.

VSMC Phenotypes	Technique	Tissue	Annotation	Marker Genes	Characteristics
Secreting VSMC [111]	scRNA-seq	3 normal controls; 8 aneurysmal aorta samples from ATAA patients;	Fibromyocyte	*ACTA2*; *MYL9*; *COL1A2*; *COL8A1*	Share properties of fibroblasts and muscle cells; Upregulate collagen, proteoglycan genes.
Secreting VSMC [112]	scRNA-seq	3 normal controls; 8 aneurysmal aorta samples from ATAA patients;	Secreting VSMCs	*ACTA2*; *MYH11*; *COL1A1*; *COL1A2*; *FABP4*	Significantly increase their percentage in AA lesions.
Secreting VSMC [113]	scRNA-seq	3 normal controls; 8 aneurysmal aorta samples from ATAA patients;	Myofibroblast	*ACTA2*; *TAGLN*; *DCN*; *LUM*; *VCAN*; *CLU*; *FN1*; *LTBP2*; *CTGF*; *AEBP1*	Share characteristics of VSMCs and fibroblasts; Characterized by ECM organization, proliferation and migration.
Secreting VSMC [114]	scRNA-seq	WT and *Yap1^f/f^*; *Myh11*-*CreER^T2^* mice; Challenged by HFD and AngII	ECM-producing VSMCs	Not discussed	Not discussed
Proliferating VSMC [111]	scRNA-seq	3 normal controls; 8 aneurysmal aorta samples from ATAA patients;	Proliferating VSMC1	*TPM2*; *MAP1B*; *MYH11*; *CCND1*; *CALD1*; *MYH10*; *TPM4*; *FGF1*; *SPARC*; *FTH1*	Upregulate cyclin gene expression; Downregulate cell–cell junction; Express high level of synthetic VSMC marker genes; Express low levels of collagen and proteoglycan genes.
			Proliferating VSMC2	*GAS6*; *IGFBP2*; *MGP*; *FTH1*; *SPARC*; *FGF1*; *TPM4*; *MYH10*; *CCND1*; *TPM2*	Same as above
Proliferating VSMC [115]	Lineage tracing	*Apoe^−/−^* (*Apoe^−/−^*; *Myh11CreERT2^T2^;mT/mG^f/f^*) mice; TGFβR2^iSMC^ (*Myh11CreER^T2^;mT/mG^f/f^;Tgfbr2^f/f^*) mice; Challenged by HCHFD;	MSC-like VSMCs	*CD105*; *CD73*; *CD90*; *CD44*; *Sca*-*1*	Monoclonal expansion of lineage-traced VSMCs was found in 9/10 samples.
Proliferating VSMC [116]	Lineage tracing	*Myh11-CreER^t2^/Rosa26-Confetti* mice; Challenged by AngII and anti-TGF;	/	/	Monochromatic patches of VSMCs were found in the medial layer of 5 out of the 6 animals analyzed.
Mesenchymal VSMC [112]	scRNA-seq	3 normal controls; 8 aneurysmal aorta samples from ATAA patients;	Msenchymal-like VSMCs	*ACTA2*; *MYH11*; *CD34*; *PDGFRA*	Play a poor role in the communication network with immune cells.
Immune-related VSMC [112]	scRNA-seq	3 normal controls; 8 aneurysmal aorta samples from ATAA patients;	Macrophage-like VSMCs	*ACTA2*; *MYH11*; *CD14*; *CD68*	Communicate with all immune cells and sent maximum number of the signaling pathways.
			T-cell-like VSMCs	*ACTA2*; *MYH11*; *CD3D*; *CD3G*	Act as a signal sender in the communication network with immune cells.
Immune-related VSMC [111]	scRNA-seq	3 normal controls; 8 aneurysmal aorta samples from ATAA patients;	Inflammatory1	*CXCR4*; *CCL4*; *CXCL12*; *TNF; CCL20*; *IFNG*; *XCL1*; *XCL2*; *LTB*; *CCL5*; *TNFSF9*; *IL32*	Show a T lymphocyte–like gene expression profile.
			Inflammatory2	*C1QA*; *C1QB*; *IL1RN*; *IL1B*; *IL18*; *CCL3*; *CCL4L2*; *CCL3L1*; *CXCL1*; *CXCL3*; *CXCL8*; *CXCL16*	Characterized by expression of macrophage markers.
			Inflammatory3	*IFIT1*; *IFI6*; *CXCL9*; *CXCL10*; *CCL28*; *IL15*	Expressed interferon-induced gene.
Immune-related VSMC [117]	scRNA-seq	4 healthy controls; 5 diseased aortas from ATAD patients;	Monocyte-like VSMCs	*CD93*; *THBD*	Characterized by expression of monocyte markers.
Immune-related VSMC [115]	Lineage tracing	*Apoe^−/−^* (*Apoe^−/−^*; *Myh11CreERT2^T2^;mT/mG^f/f^*) mice; TGFβR2^iSMC^ (*Myh11CreER^T2^;mT/mG^f/f^;Tgfbr2^f/f^*) mice; Challenged by HCHFD;	Macrophage-like cells	*Mac2*	Characterized by expression of macrophage markers.
Proinflammatory VSMC [118]	scRNA-seq	C57BL/6J mice; Challenged by elastase;	Inflammatory-like SMCs	*Mac2*; *Sparcl1*; *Igfbp5*; *Sncg*; *Thbs1*; *Notch3*; *Cd68*; *Cxcl1*; *Cxcl2*; *Il1r1*	Displayed a pro-inflammatory phenotype; Increased at late stage of AAA development.
Senescent VSMC [117]	scRNA-seq	4 healthy controls; 5 diseased aortas from ATAD patients;	VSMC8	*MT1G*; *MT1M*	Express metallothionein superfamily genes; Have minimum differentiation potential.
Senescent VSMC [118]	scRNA-seq	C57BL/6J mice; Challenged by elastase;	Dedifferentiated SMCs	*Ifrd1*; *Klf4*; *gata6*; *Mt1*; *Mt2*; *Hk2*;	Show the lowest expression of contractile markers; Have altered metabolic profile to glycolysis.
Stressed VSMC [111]	scRNA-seq	3 normal controls; 8 aneurysmal aorta samples from ATAA patients;	Stressed VSMCs	*FOS*; *ATF3*; *JUN*; *HSPB8*; *ACTC1*; *BRD2*; *KLF10*; *DUSP1*;	Share several features with contractile VSMCs, except the activation of stress response genes.
Stressed VSMC [118]	scRNA-seq	C57BL/6J mice; Challenged by elastase	Proliferative VSMCs	*Fos*; *Jun*; *Klf2*; *Klf4*; *gata6*; *Atf3*	Upregulate expression of genes associated with stress; Gain an inflammation-activated profile.

Vascular smooth muscle cell (VSMC), aortic aneurysm (AA), ascending thoracic aortic aneurysm (ATAA), single-cell RNA sequencing (scRNA-seq), high fat diet (HFD), mesenchymal stem cell (MSC), high cholesterol high fat diet (HCHFD), Angiotensin II (AngII), acute thoracic aortic dissection (ATAD).

## Data Availability

The authors confirm that this is a review of the literature and there is no primary data associated with this manuscript.

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
