# Peer review of "Smooth Muscle Heterogeneity and Plasticity in Health and Aortic Aneurysmal Disease"

_ijms, 2023, doi:10.3390/ijms241411701_

Round 1

Reviewer 1 Report

The information is presented in a very detailed fashion and precisely described from a very high number of publications.

More integration and summarizing paragraphs of information from selected sources would significantly improve the quality of submission and practical application of presented knowledge.

The reduction in the number of pages and integration of information based on the scientific findings (that are related to different clinical forms of AAs) would be highly beneficial for any reader.

The information is presented in a very detailed fashion and precisely described from a very high number of publications.

More integration and summarizing paragraphs of information from selected sources would significantly improve the quality of submission and practical application of presented knowledge.

The reduction in the number of pages and integration of information based on the scientific findings (that are related to different clinical forms of AAs) would be highly beneficial for any reader.

Author Response

Response to Reviewer1:

We sincerely appreciate your attention to the details of the manuscript's structure. Based on your suggestions, we have carefully re-read the manuscript and specifically integrated the content of Part 2 and Part 3 to better convey our intended scientific findings. Additionally, we have further refined the sentences to make the paper more concise and fluid. These modifications will aid readers in better understanding and comprehending our review.

Reviewer 2 Report

The presented manuscript is a synthesis of research and a thorough assessment of it. The conclusions enable a broader look at the presented field of knowledge and set directions for further research.

Author Response

Response to Reviewer 2:

We sincerely appreciate your comments for this review paper.

Reviewer 3 Report

Authors,

Please provide some information about epigenetic mechanisms (ie. miRNA) involved in pathophysiology of VSMC in context of aortic aneurysm and aortic occlusive disease. 

Author Response

Response to Reviewer3:

We sincerely appreciate your attention to the contents of the manuscript. We acknowledge the significant role of epigenetic regulatory mechanisms, such as miRNA, in driving phenotypic changes in VSMCs and their potential association with VSMC heterogeneity. In accordance with your suggestion, we have included a discussion on this topic in Part 5. It is important to emphasize that evidence supporting epigenetic dysregulation is likely to contribute to VSMC dysfunction and the development of vascular diseases. However, further evidence is needed to substantiate this view, and there is currently a lack of direct comparative studies on epigenetic heterogeneity among different regions of VSMCs.

Reviewer 4 Report

The review by Yunwen Hu, Zhaohua Cai* and Ben He* Smooth muscle heterogeneity and plasticity in health and aortic aneurysmal disease is rather interesting and important. In general I have only a few suggestions.

1. In Figure one, there is no indication that the aortic intima of large animals and humans consists of not only endothelial cells, but also contain smooth muscle cells and pericytes (10.3390/ijms23042152 ).

2. It could be interesting to know the origin of these SMCs.

3. Although there is rather detailed analysis of the phenotypes. however, the analysis of these phenotypes at the ultrastructural level is missing. If this information does not exist it is necessary to stress this.

4. Also, impaired functional phenotypes could be important to illustrate or at least to indicate where one could find this information if it exists.

 5. In Figure 1, there is no indication what green and blue lines represent. In one case these lines indicate elastic membranes (1B) and in another possibly collagen (1C).

6. Also, microfibrils could be from collagen and from elastin.

7. The role of fibrillin is not clear.

Author Response

Response to Reviewer4:

Thank you for your careful review of our manuscript and figures, and for providing valuable suggestions. We greatly appreciate your guidance. Here is our response to your suggestions:

Response to Suggest 1:

Regarding the question of whether pericytes are present in the intima, we have read the reference you provided. In the high-quality review you provided, it was mentioned that most endothelial cells express VSMC and pericyte antigens. However, we hold the point that the expression of VSMC and pericyte antigens alone is not sufficient to prove the presence of pericytes in the intima. Further isolation and functional validation of these cells are required. In addition, these results are primarily observed in atherosclerotic lesions. Based on our current knowledge and experience, we tend to believe that pericytes are not present in the intima of normal blood vessels. Therefore, we have decided not to show pericytes in Figure 1.

Response to Suggest 3 and Suggest 4:

The impaired functional phenotypes are summarized from currently available sc-RNAseq and lineage tracing evidence, and we currently lack ultrastructural analysis of these phenotypes due to the lack of methods to isolate these specific subpopulations of cells. In Part 6 of the revised manuscript, we have added that sc-RNAseq only provides transient transcriptional information, and the observed phenotypes may only reflect the transient cellular states. Therefore, we believe that an important and urgent task is to attempt the isolation and stable cultivation of VSMC subpopulations discovered by sc-RNAseq, such as Sca1+ VSMCs, as this will provide direct evidence for the diverse heterogeneity of VSMCs.

Response to Suggest 5 and Suggest 6:

Regarding the lines in Figure 1B, we confirm that they represent elastin membranes, while the green lines in Figure 1C indicate elastin microfibrils. In the middle panel of Figure 1C, we already highlighted the function of fibrillin-1: it mediates the connection between integrin and elastic microfibrils, thereby anchoring VSMCs to the elastin membrane. Additionally, fibrillin-1 is believed to be associated with latent transforming growth factor β binding proteins (LTBPs), participating in the regulation of TGF-β concentration in the extracellular matrix. In the revised figure, we have decided to add a separate legend in Figure 1C. This modification will help readers to better understand the information conveyed in the figure.

Reviewer 5 Report

In the manuscript entitled "Smooth muscle heterogeneity and plasticity in health and aortic aneurysmal disease" the Authors discussed about the heterogeneity and diversity of VSMCs in relationship to aortic aneurysms (genetic and sporadic AAs). The topic is interesting and the review well-written. However, since the review focused on different biomolecular aspects, it could be appropriate adding a paragraph about the role of miRNAs in genetic and sporadic AAs and their effects on VSMC phenotype, reporting up-to-date papers (i.e Yang K et al. Eur Heart J. 2020Si X et al. Life Sci. 2022; D'Amico F et al. IJMS 2020).

Author Response

Response to Reviewer 5:

Thank you very much for your recognition of the topic of this review. Our main objective in this review is to summarize and depict the heterogeneity of VSMCs both in physiology and in the context of aortic aneurysms. Our aim is to convey the message that VSMCs exhibit a rich heterogeneity and that the conventional binary categorization into contractile VSMCs and synthetic VSMCs oversimplifies the actual scenario.

After careful consideration, we have included in Part 5 a discussion of the role of epigenetic mechanisms in regulating VSMC phenotypes, as well as the references you suggested.

Reviewer 6 Report

In this interesting review, the authors discuss the Smooth muscle heterogeneity and plasticity in health and aortic aneurysmal disease.

First, they address the detailed description of the aorta and the role of VSMCs in physiology and in pathophysiological aneurysmal aortas.

In further parts, they introduce VSMC heterogeneity in normal aorta and then VSMC phenotypic diversity in hereditary AAs. In the fourth part the authors present VSMC phenotypic diversity in non-hereditary AAs.

Finally, they discuss the role of VSMC phenotypic diversity in AAs and conclude the review with a future perspective.

This is a very well written review, which gives us a bright overview over the relationship between VSMCs and aortic aneurysmal disease.

I have only one minor remark, which I believe is important for the completion of this review.

Table 1 should be optimized since the lettering is mainly shifted and there should be an extra column for the cited literature and publication year, which would be easier to read the information presented in this table.

Author Response

Response to Reviewer6:

Thank you for your suggestions regarding Table 1. In the revised manuscript, we will optimize the layout of Table 1 to ensure consistent alignment of the lettering. Regarding the suggestion of adding an additional column specifically for references, after careful consideration, we have decided not to adopt it. The reason is that we already provide the citation information in the first column of Table 1, which allows readers to conveniently access the reference details. In addition, having too many columns in the table may prevent readers from quickly obtaining the information they need and may also create formatting difficulties.

Once again, we sincerely appreciate your review of our manuscript and figures, as well as your valuable suggestions. Your guidance has been very helpful to our research. We will revise the manuscript and figures according to your suggestions to ensure that they are clearer and easier to understand.

Round 2

Reviewer 4 Report

Comments

The review by Yunwen Hu, Zhaohua Cai and Ben He "Smooth muscle heterogeneity and plasticity in health and aortic aneurysmal disease" is rather interesting and important. In general I have only a few suggestions.

1. In Figure 1B, there is no indication that the aortic intima of large animals and humans consists of not only endothelial cells, but also contain smooth muscle cells and pericytes (10.3390/ijms23042152 ).

The authors did not correct this item (see Figure I below) from all these images taken from internet it is clear that intima consists not only from endothelial cells as it is shown in Fig. 1 by the authors. In aorta intima is composed of several cell layers. This should be indicated in Figure. One could discuss the origin of these cells. However, it is not possible to deny the existence of other non-endothelial cells in intima.

Figure I

2. It could be interesting to know the origin of these SMCs

(and how it was proved).

No enough correction

3. Although there is rather detailed analysis of the phenotypes. however, the analysis of these phenotypes at the ultrastructural level is missing. If this information does not exist it is necessary to stress this.

No enough corrections

4. Also, impaired functional phenotypes could be important to illustrate or at least to indicate where one could find this information if it exists.

No enough corrections

 5. In Figure 1, there is no indication what green and blue lines represent. In one case these lines indicate elastic membranes (1B) and in another possibly collagen (1C).

6. Also, microfibrils could be from collagen and from elastin.

This should be clear indicated in the text.

7. The role of fibrillin is not clear.

This item is not discussed at all.

Now the text is completely unreadable due to the use red letters a. New part should be indicated better. It is not necessary to show in such a way the text eliminated. My suggestion new major revision or rejection.

Author Response

Reply to Suggest 1:

We appreciate your careful attention to Figure 1 and your views on the composition of aortic endothelial cells. We have carefully read the references you provided and have revised Fig 1 B. We acknowledge that endothelial injury as a prerequisite process for arterial clamp formation was overlooked in the previous revised version. We recognize that turbulent blood flow in the ascending aorta leads to endothelial cell injury and provides an opportunity for infiltration of inflammatory cells, VSMCs into the endothelial layer. Therefore, in the new version of Figure 1B, we have added inflammatory cells and deformed VSMCs at the initiation site of aortic wall tears to succinctly characterize the complex cellular composition of endothelial injury. These changes not only make Fig1B more rigorous, but also enrich our field of knowledge!

Reply to Suggestion 2:

Thanks to your suggestion, we have indeed shown the embryologic origin of VSMCs in the aortic wall in Figure 1 A. Nevertheless, in the second part of this paper, we briefly summarized the embryologic origins of VSMCs in different aortic segments. For example, VSMCs in the aortic root are mainly derived from the origin of the SHF (second heart field). In addition, our Part 2 summarizes the latest findings on the distribution characteristics of CNC-derived VSMCs and SHF-derived VSMCs in the ascending aorta and aortic root. lineage-tracing is the primary method for verifying the embryologic origin of VSMCs, and Mark W. Majesky has written a detailed review in this area. (10.1161/ATVBAHA.107.141069).

 Reply to Suggest 3:

Thank you for your interest in the ultrastructural characterization of VSMCs. There is indeed a lack of relevant information at present, which we have highlighted in Part 6 of the revised version. We believe that an important future research direction is to isolate phenotypes of VSMCs with stable characteristics from the vessel wall and provide a comprehensive summary of their ultrastructure. Such studies will provide more direct evidence for the existence of different phenotypes of VSMCs and lay the foundation for the construction of cellular models that will help to further understand the role of VSMCs in aneurysm formation and development. 

Reply to Suggest 4:

Thank you for your comment. In fact, in Table 1, we summarize the functional phenotypes of VSMCs and provide the reference sources in our "VSMC phenotypes" column. Based on your suggestion, to help readers find the original literature more quickly, we decided to add the reference after each functional phenotype in the revised version of Figure 2 as well.

Reply to Suggest 5,6,7:

Thank you again for your careful review of Fig1. In response to your feedback, we have revised Fig1B and Fig1C to indicate what the green and blue lines refer more clearly. In Figure 1C, we have further clarified that microfibrils represent extensions of elastic fibers and have also changed the annotation to convey this accurately. We recognize that in the previous version of Figure 1C we confused the concepts of integrins and fibrillin. In the revised figure, we have changed the legend of Figure 1 to clarify the role of fibrillin.

Reviewer 5 Report

No further comments.

Round 3

Reviewer 4 Report

I agree that the paper can be published in its current form.